# Metal–Organic Frameworks for Overcoming the Blood–Brain Barrier in the Treatment of Brain Diseases: A Review

**DOI:** 10.3390/nano14171379

**Published:** 2024-08-23

**Authors:** Hafezeh Nabipour, Sohrab Rohani

**Affiliations:** Department of Chemical and Biochemical Engineering, University of Western Ontario, London, ON N6A 5B9, Canada; hnabipou@uwo.ca

**Keywords:** metal–organic frameworks, delivery systems, blood–brain barrier, brain diseases

## Abstract

The blood–brain barrier (BBB) plays a vital role in safeguarding the central nervous system by selectively controlling the movement of substances between the bloodstream and the brain, presenting a substantial obstacle for the administration of therapeutic agents to the brain. Recent breakthroughs in nanoparticle-based delivery systems, particularly metal–organic frameworks (MOFs), provide promising solutions for addressing the BBB. MOFs have become valuable tools in delivering medications to the brain with their ability to efficiently load drugs, release them over time, and modify their surface properties. This review focuses on the recent advancements in molecular-based approaches for treating brain disorders, such as glioblastoma multiforme, stroke, Parkinson’s disease, and Alzheimer’s disease. This paper highlights the significant impact of MOFs in overcoming the shortcomings of conventional brain drug delivery techniques and provides valuable insights for future research in the field of neurotherapeutics.

## 1. Introduction

The global burden of brain diseases is on the rise, significantly contributing to increasing disability and mortality rates. This growing concern stems from various factors, including an aging population, increased life expectancy, and the widespread prevalence of risk factors. As populations age, the incidence of neurodegenerative disorders like Alzheimer’s and Parkinson’s diseases, along with cerebrovascular conditions such as stroke, has surged, presenting formidable challenges to healthcare systems and societies. Brain tumors, whether primary or metastatic, remain among the most challenging neurological conditions, characterized by high mortality rates and significant morbidity, despite advancements in medical technology and treatments. The intricate structure and function of the brain complicate surgical interventions, while the blood–brain barrier (BBB) presents significant obstacles to the effective delivery of chemotherapy drugs. Additionally, traumatic brain injuries and spinal cord injuries, often resulting from accidents, continue to be major contributors to global disability and mortality, impacting cognitive abilities, physical mobility, and independence. Moreover, increasing attention has been paid to rare brain diseases like amyotrophic lateral sclerosis (ALS). Although rare, these conditions collectively affect a substantial number of people worldwide. Brain diseases or disorders represent a major global health issue, significantly impacting individuals’ quality of life and their ability to function independently. Recent advancements in neuroscience research are shedding light on the root causes of brain diseases, leading to better diagnostic techniques, more effective treatments, and preventive measures to address this major public health issue. These disorders can affect children, resulting in intellectual disability and mobility impairments, while in adults, they can present as depression, memory loss, specific neurological disorders, brain tumors, and brain transport issues. Both genetic factors and various environmental conditions, including aging and stress, can contribute to the development of these diseases before birth and during brain development. The journey to create effective treatments for brain diseases is fraught with challenges, including long research and development timelines and a high rate of failures during clinical trials. A significant hurdle is the effective delivery of medications to the brain, hindered by biological barriers such as the BBB [1,2,3,4]. To address these challenges, researchers are exploring innovative drug delivery methods and treatment strategies. These include nanosystems containing biomolecules in the form of liposomes [5,6,7], micro- and nano-emulsions [8,9,10], polymeric micelles [11,12], nanosuspensions [13,14], niosomes [15,16], and nanoparticles (NPs) [17,18], offering promising solutions to enhance drug delivery to the brain.

Previous studies have demonstrated that metal–organic frameworks (MOFs), an advanced type of porous hybrid organic–inorganic materials [19,20], hold significant promise as brain-targeting nanocarriers [21,22]. MOFs are a class of advanced materials characterized by their crystalline structure, and they are formed by the coordination of metal ions or clusters with organic ligands [19,20]. MOFs have applications in energy storage [23], catalysis [24], gas adsorption and separation [25], supercapacitors [26], the removal of heavy metals [27], membranes [28], biomedicine [29], and more. MOFs demonstrate significant promise for drug delivery due to their unique advantages, including a large surface area, high porosity, adjustable pore sizes, surface functionalization, substantial drug loading capacity, protective encapsulation, versatility, low density, structural diversity, and improved thermal and chemical stability. Nonetheless, they also present some drawbacks, as outlined in Table 1. Similarly, while liposomes, micelles, and polymeric nanoparticles have been employed in drug delivery, they have their own set of advantages and disadvantages, as detailed in Table 1 [21,22,29,30,31]. A comparative analysis of these systems in terms of efficacy, safety, and scalability, highlighting their respective strengths and limitations, is provided in Table 1 for a clearer understanding.

Despite the advantages and disadvantages reported for MOFs, they have been widely utilized in diagnosing and treating various cancers and brain diseases due to their unique properties. This review explores the application of MOFs as drug delivery systems for addressing cancer and neurological disorders. It examines the characteristics, synthesis, and detection of MOFs, focusings particularly on their potential for treating brain conditions such as stroke, glioblastoma multiforme, Parkinson’s disease, and Alzheimer’s disease. This review also discusses the challenges in and strategies for overcoming the BBB to improve drug delivery to the brain.

## 2. Synthesis and Characterization of MOFs

The production of MOFs is influenced by numerous factors, including the reaction time, temperature, solvent type, metal ions, organic linkers, and crystallization rate, all of which significantly impact the structure and properties of the final product. Various methods can be used to synthesize MOFs, each offering unique advantages and challenges. These methods include conventional hydrothermal/solvothermal synthesis, microfluidic methods, microwave-assisted synthesis, sonochemical methods, reverse-phase microemulsions, dry gel conversion, ionothermal synthesis, diffusion synthesis, electrochemical synthesis, and mechanochemical methods. Each technique provides different reaction conditions and energy inputs, allowing researchers to tailor the synthesis process to achieve desired MOF properties (Figure 1).

Understanding the structure, properties, and potential uses of MOFs requires a comprehensive characterization of the final product. This process involves employing a variety of analytical techniques to uncover both the physical and chemical attributes of MOFs. Powder X-ray diffraction (PXRD) and single-crystal X-ray diffraction (SCXRD) are used to determine crystal structures. Infrared (IR) and Raman spectroscopy help identify molecular vibrations. The Brunauer–Emmett–Teller (BET) method measures the surface area, while thermogravimetric analysis (TGA) and differential scanning calorimetry (DSC) assess thermal stability. Inductively coupled plasma mass spectrometry (ICP-MS) provides the elemental composition, and X-ray photoelectron spectroscopy (XPS) gives insights into the surface chemistry. Nuclear magnetic resonance (NMR) spectroscopy reveals dynamic and structural information for a number of nuclear species, while scanning electron microscopy (SEM) and transmission electron microscopy (TEM) offer detailed images of the morphology and internal structure of MOFs (Figure 1) [20,21,22,23,24,25,26,27,28,29].

## 3. Drug Loading and Drug Release in MOFs

MOFs represent a groundbreaking class of materials in brain delivery, offering unique advantages due to their exceptional structural properties and versatility. Before using MOFs for drug delivery, it is crucial to thoroughly assess their biocompatibility. The careful selection of metal ions and organic linkers in MOF assembly is essential to ensure safety and minimize toxicity. Many MOFs incorporate essential metals, such as iron, zinc, copper, and magnesium, which are naturally present in the human body. Additionally, metals like zinc, iron, magnesium, calcium, titanium, and zirconium are considered suitable for biocompatible MOFs due to their low toxicity profiles. The choice of organic linkers is equally important for the overall biocompatibility of MOFs. Ideally, these linkers should be easily excreted or metabolized after in vivo application to avoid toxic buildup. Trimesic acid, terephthalic acid, 2,6-naphthalenedicarboxylic acid, and imidazolate linkers have shown higher biocompatibility due to their ability to be readily removed under physiological conditions [32,33].

It should be highlighted that the intrinsic limitations of many pharmaceutical compounds, such as poor tissue penetration, instability, and short serum half-life, have driven researchers to explore innovative solutions. MOFs as carriers have been specifically designed to address these challenges, aiming to enhance immune escape, facilitate entry into tumor cells, and improve the overall efficiency of pharmacologically active biomolecules [34]. MOFs can enhance the drug loading capacity due to their large surface areas and tunable pore sizes, allowing for increased concentrations of active compounds to be delivered to the target sites. Loading pharmacologically active molecules in MOFs is achieved through various methods, with each exploiting the unique structural properties of these materials. Surface adsorption is a highly effective method for drug loading in MOFs, utilizing their high porosity and large surface area. Drugs are loaded onto MOF surfaces through weak interactions like van der Waals forces, dipole interactions, and hydrogen bonding. These versatile techniques allow for a broad range of drug–MOF combinations without stringent pore size requirements. The surface properties and modifications of MOFs are crucial for optimizing drug loading and release. Solvent effects also significantly influence the adsorption process, emphasizing their importance in MOF-based drug delivery system designs [35].

Nadizadeh et al. successfully loaded ibuprofen (IBU) into two MOFs ({Cu_2_(1,4-bdc)_2_(dabco)}_n_ and {Cu_2_(1,4-bdcNH_2_)_2_(dabco)}_n_) using a mechanochemical method, achieving loading efficiencies of approximately 50.54% and 50.27%, respectively [36]. The one-pot method for drug loading in MOFs is an efficient and cost-effective technique that involves the simultaneous synthesis of the MOF and incorporation of drug molecules. This approach ensures a homogeneous distribution of the drug within MOF pores, effectively overcoming the limitations posed by small pore sizes. The method streamlines the process, reducing the reaction time and waste production while achieving a high drug loading capacity [37]. An example is the incorporation of doxorubicin (DOX), an anticancer drug, into zeolite imidazole framework-8 (ZIF-8) during its synthesis. This one-pot synthesis/loading process resulted in efficient drug delivery vehicles for cancer therapy with loadings of up to 20% [38]. A commonly used method for incorporating drugs into MOFs is the impregnation technique. This method involves immersing the MOF in a concentrated drug solution, which allows the drug molecules to diffuse into the MOF’s pores. This technique is especially effective for small-molecule drugs that can easily penetrate the MOF structure. The main interactions between the drugs and MOFs usually include hydrogen bonding, π-π interactions, and van der Waals forces [39]. Liu et al. demonstrated the successful encapsulation of the poorly water-soluble analgesic Fenbufen (FBF) using the impregnation method. They introduced cyclodextrin-based MOFs (CD-MOFs) into an ethanol solution of FBF and allowed the reaction to proceed over 24 h, achieving a significant loading of FBF of up to 196 mg/g [40,41]. Post-synthetic modification is a powerful strategy for loading drugs into MOFs, involving the attachment of drug molecules to the MOF surface or within its pores after initial synthesis. This approach starts with the preparation and isolation of MOF nanoparticles, ensuring they meet the required size, shape, and physicochemical characteristics. Drugs can then be incorporated through various mechanisms, such as forming coordination bonds with metal nodes, establishing covalent connections with linker functional sites, or adsorbing onto the exterior surfaces of pre-synthesized MOF particles. This versatile method allows for precise control over drug loading, enhancing the efficacy and functionality of MOF-based drug delivery systems [42]. For instance, ZIF-8 was used for 5-fluorouracil (5-FU) delivery, achieving a notable capacity of around 660 mg of 5-FU/g of ZIF-8 through post-synthetic modification [43]. Another method of loading is covalent binding. The surfaces of MOFs are characterized by a variety of functional groups, such as amino, carboxyl, and hydroxyl moieties. These active sites create valuable opportunities for forming robust covalent bonds with complementary functional groups on target molecules. Utilizing these strong chemical interactions allows for a more stable and controlled incorporation of functional molecules within MOF structures. This approach not only enhances the stability of the incorporated molecules but also offers greater control over their release kinetics. The transition to covalent binding strategies in MOF functionalization marks a significant advancement, providing a more reliable and efficient method for integrating functional molecules and addressing the limitations of weaker interaction forces. [44,45]. For example, Morris et al. developed UiO-66-N_3_ nanostructures and utilized a strain-promoted click reaction to covalently functionalize oligonucleotides onto the MOF surface, effectively reducing issues related to drug leaching [46].

After drug loading, understanding drug release and its mechanisms is crucial for effective delivery to brain. The release of drugs from MOFs can be tailored through various mechanisms to achieve controlled and targeted delivery. One common mechanism is diffusion-controlled release, where drug molecules exit MOF pores due to concentration gradients, with the rate being influenced by factors like pore size and drug–MOF interactions [47]. In some cases, drug release is triggered by the degradation of the MOF under specific physiological conditions, such as changes in the pH redox environment, or enzymatic activity [48].

Additionally, MOFs can be engineered for responsive drug release based on various stimuli, including pH, temperature, light, or magnetic fields. For instance, pH-responsive MOFs can release drugs more rapidly in acidic environments. An example is Zn-GA (GA: L-glutamic acid), a biocompatible MOF developed by Lin and colleagues, which was capable of loading methotrexate (MTX) with a capacity of 12.85 wt.% when heated to 80 °C, demonstrating both heat- and pH-responsive drug release. At a pH of 5.0, the release rate was 43%, increasing to 68% at higher temperatures [49]. Other mechanisms include the breaking of covalent bonds between the drug and the MOF structure under specific conditions, such as acid-labile bond hydrolysis at a low pH [48]. Sun et al. developed a drug delivery system with 5-FU loaded into ZIF-8, achieving a high drug loading capacity of up to 60%. This system exhibited controlled drug release, with 5-FU releasing more quickly at pH 5.0 compared to pH 7.4 [50]. Specifically, over 45% of the drug was released within the first hour at pH 5.0, while only 17% was released at pH 7.4. Another mechanism involves the collapse or disassembly of MOFs under certain conditions, leading to rapid drug release. This can be triggered by factors such as pH changes or competitive ion binding [48]. The si-RNA@CCM-loaded-MIL-101-NH_2_ system, which encapsulates curcumin (CCM) and si-RNA within MIL-101-NH_2_, effectively protects both CCM and si-RNA from nuclease degradation and facilitates lysosomal escape. In acidic environments, MIL-101-NH_2_ begins to disintegrate, releasing the encapsulated CCM [51]. The choice of release mechanism depends on the MOF’s structure, the drug’s properties, and the intended release profile. For example, studies have shown that pH-responsive MOFs can release drugs more rapidly in acidic conditions, which is beneficial for targeted delivery in environments containing cancerous tumors. The versatility of MOFs in drug release mechanisms allows for precise control over drug delivery, making them promising platforms for advanced therapeutic applications [52].

## 4. MOFs in the Treatment of Brain Diseases

The BBB is an extraordinary biological structure that acts as a protective barrier between the central nervous system and the bloodstream. This highly specialized barrier regulates the movement of chemicals from the bloodstream into neural tissue, which is crucial for maintaining brain homeostasis. The BBB’s primary components are tightly connected to brain endothelial cells that form a continuous layer lining cerebral blood vessels. These cells are characterized by their unique properties, including tight junctions, a lack of fenestration, and minimal pinocytotic activity, which contribute to the BBB’s highly selective nature. The BBB’s complex structure and composition allow it to effectively filter out potentially harmful substances while permitting the passage of essential nutrients and molecules necessary for brain function. This selective permeability is further enhanced by the presence of other cellular components, such as pericytes, astrocytes, and the basement membrane, which collectively form the neurovascular unit. These additional elements work in concert with endothelial cells to maintain the integrity and functionality of the BBB. The BBB is essential for maintaining brain health but presents significant challenges in treating neurological disorders. The BBB regulates the entry of vital substances such as glucose, amino acids, neurotransmitters, and ions into the brain via various membrane transporters, pumps, and receptor-mediated endocytosis for specific molecules. Transporters in the endothelial cell membrane act as a barrier against the infiltration of foreign compounds and pharmaceuticals, protecting the brain from harmful substances. However, in many neurological conditions, the structural and functional integrity of BBB cells can be compromised, leading to altered permeability. This change can cause the accumulation of water and plasma proteins within the central nervous system, potentially increasing brain volume and intracranial pressure, which may result in edema and damage to the myelin sheath. For example, in Alzheimer’s disease, the endothelial barrier becomes more permeable, allowing amyloid-beta plaques to accumulate in the brain parenchyma. Similarly, in Parkinson’s disease, the BBB’s integrity is reduced and its permeability is increased due to the buildup of alpha-synuclein at the axon terminals of dopamine neurons.

Many therapeutic agents are unable to cross this barrier, limiting their effectiveness in treating brain-related conditions. This obstacle has spurred the development of innovative strategies to overcome the BBB, with nanoparticle-based delivery systems emerging as a promising approach. Nanoparticles offer a unique solution to the challenge of drug delivery across the BBB. As shown in Figure 2, these tiny particles can be designed to interact with the BBB in various ways, such as paracellular transport, carrier-mediated transport, and cell-mediated transport, enabling them to transport therapeutic agents into the brain. Some nanoparticles utilize adsorptive-mediated transcytosis, where their positive charge interacts with the negatively charged cell membrane of brain endothelial cells, facilitating uptake and transport. Additionally, others employ receptor-mediated transcytosis, using specific ligands to bind to receptors on the BBB and trigger internalization. The use of nanoparticles for brain drug delivery offers several advantages over traditional methods. These include improved drug stability, targeted delivery to specific brain regions or cell types, and enhanced BBB penetration [53,54,55,56].

MOFs have attracted significant interest in recent years due to their potential to overcome the challenges posed by the BBB in treating brain disorders. This interest is due to their unique physicochemical properties, including high drug loading efficiency, controlled drug release, nanoscale size, ease of surface modification, stability, biodegradability, and biocompatibility with low systemic toxicity. A key advantage of MOFs in crossing the BBB is their ability to be functionalized with various ligands and molecules that can target specific receptors on the BBB. This receptor-mediated transport mechanism improves the ability of MOFs to penetrate the BBB and deliver therapeutic agents directly to brain tissues. Additionally, modifying the surface of MOFs can enhance their biocompatibility, stability, and targeting capabilities, further increasing their potential as drug delivery systems [57,58].

## 5. MOFs in the Treatment of Glioblastoma Multiforme

Glioblastoma multiforme (GBM) is a highly aggressive brain tumor, and the BBB presents a significant challenge for drug delivery. This novel approach uses a nanocarrier inspired by the rabies virus (RABV) called MILB@LR. By emulating the bullet-shaped structure and surface functions of RABV, MILB@LR can effectively penetrate the BBB and target brain tumors. The nanocarrier is based on the iron (Fe)-based MIL family of MOFs, which offer a controllable morphology, adjustable surface modification, and a high loading capacity. It features a lipid bilayer coating with the RVG29 peptide, which binds to nicotinic acetylcholine receptors (nAchRs) that are overexpressed on both BBB and GBM cells. This RABV-mimetic design allows MILB@LR (Figure 1a) to efficiently cross the BBB and accumulate in brain tumors. Loaded with the antitumor drug oxaliplatin (OXA), MILB@LR has demonstrated impressive efficacy in inhibiting tumor growth compared to methods that only replicate isolated features of RABV. In vivo studies have shown that MILB@LR effectively reaches the brain and penetrates deeply into glioma tumors, overcoming the barrier of drug delivery across the BBB, which typically restricts more than 98% of therapeutic agents (Figure 1b) [59,60].

Researchers developed a novel copper-based nanoplatform, BSO-CAT@MOF-199@DDM (BCMD), designed to address glioblastoma multiforme (GBM) within its immunosuppressive tumor microenvironment. BCMD aimed to induce cuproptosis, a copper-triggered form of cell death, by releasing Cu^2+^ ions that were converted to toxic Cu^+^ in the slightly acidic tumor milieu, influenced by high levels of ferredoxin 1 (FDX1) in GBM cells. MOF-199, with a significant copper content of around 30%, was analyzed for its Cu^2+^ and buthionine-sulfoximine (BSO) release profiles. Cu^2+^ release from BCMD was stable at pH 7.4, with 55.3% being released over 48 h, but this value increased rapidly to 86.8% at pH 5.5. BSO, with a 6.2% loading efficiency, showed 69% release at pH 5.5 within 12 h compared to less than 40% at pH 7.4, indicating gradual degradation in acidic conditions. BCMD also released BSO and catalase (CAT) which, together, reduced glutathione synthesis and increased the oxygen content in tumor cells, enhancing susceptibility to cuproptosis. This cuproptosis induced immunogenic cell death (ICD), activating dendritic cells, increasing T-cell infiltration, and reversing the immunosuppressive tumor microenvironment. Additionally, combining BCMD with immune checkpoint blockade therapy, specifically αPD-L1 treatment, significantly improved therapeutic efficacy. BCMD also addressed the challenge of the BBB through intranasal administration, allowing direct delivery to brain tumors. While BCMD showed promising results, further research was needed to optimize dosing, assess long-term toxicity, and explore metabolic mechanisms before progressing to large animal models. This innovative nanoplatform represented a potential breakthrough in GBM treatment and other tumors with high FDX1 expression, potentially revolutionizing brain disease therapies [61].

In a groundbreaking study conducted by Pulvirenti et al., researchers developed an innovative hybrid nanosystem by integrating iron-based MILs with iron oxide magnetic nanoparticles (MNPs) to enhance the treatment of glioblastoma GBM, an extremely aggressive brain tumor. The synthesis of these hybrids involved two distinct routes: one utilizing Fe_3_O_4_ nanoparticles as the sole source of Fe^3+^ ions and another incorporating an additional external source of Fe^3+^ ions (FeCl_3_). This approach aimed to combine the magnetic properties of Fe_3_O_4_ with the high surface area and porosity of MILs, optimizing both magnetic separation and drug loading capabilities. The researchers modified Fe_3_O_4_ magnetic nanoparticles with MIL-101(Fe) to encapsulate and release temozolomide (TMZ), a chemotherapy drug used in GBM treatment. The synthesized nanocomposite demonstrated a TMZ loading capacity of 12 mg/g. TMZ undergoes hydrolysis at physiological pH to produce 5-(3-methyltriazen-1-yl)imidazole-4-carboxamide (MTIC), which further degrades rapidly into methyl diazonium ion and 5-aminoimidazole-4-carboxamide (AIC). Unlike TMZ, MTIC faces challenges in penetrating the BBB effectively. A thermogravimetric analysis estimated the MIL content in the hybrids to be between 3.6% and 5.7% depending on the synthesis route. The hybrid nanoparticles’ ability to load and release guest molecules was evaluated using Rhodamine B and TMZ, with the results demonstrating that the MNPs@MIL[b] system, which included FeCl_3_ in the synthesis, exhibited superior loading and release properties compared to MNPs@MIL[a] and bare MNPs. In vitro studies using human glioblastoma A-172 cells indicated that the internalization of MNPs@MIL[b] was more efficient than that of bare MNPs, with increased cellular uptake being observed at all tested concentrations. The cytotoxicity of TMZ-loaded MNPs@MIL[b] was significantly higher than that of free TMZ or bare MNPs, with a notable reduction in cell viability. This enhanced efficacy is particularly important given TMZ’s short half-life, which typically requires high doses to achieve the desired anticancer effect, often leading to side effects such as myelosuppression. The MIL-modified system demonstrated improved internalization in the nucleus and cytoplasm compared to unmodified magnetic nanoparticles, offering a promising approach for developing advanced drug delivery systems with enhanced loading, release, and therapeutic efficacy. This research highlights the successful integration of MIL frameworks with magnetic nanoparticles, potentially revolutionizing targeted drug delivery for GBM treatment and opening new avenues for combating this aggressive form of brain cancer [62].

In a study by Sharma and colleagues, the use of a magnesium gallate (Mg-GA) MOF for the delivery of cannabidiol (CBD) and gallic acid (GA) as potential anticancer agents was investigated. The CBD/Mg-GA MOF was characterized using various techniques, including XRD and IR spectroscopy, which confirmed the encapsulation of CBD within the MOF’s pores. The release of GA, magnesium ions, and CBD from the MOF under different pH conditions was assessed, revealing that the MOF degraded more rapidly and released its contents more readily in acidic environments, which is beneficial for targeting tumor cells. The CBD/Mg-GA MOF demonstrated significant antioxidant activity in DPPH assays, comparable to that of ascorbic acid. Hemocompatibility tests showed low hemolysis, indicating the MOF’s potential for biomedical applications. Cytotoxicity studies revealed that the CBD/Mg-GA MOF was more effective than free GA or CBD in reducing glioblastoma cell viability, with the greatest effects being observed after 48 h of treatment. The MOF also improved cellular uptake compared to free GA due to its lipophilicity. Studies on BBB permeability showed that the CBD/Mg-GA MOF increased BBB permeability, which is essential for drug delivery to the brain. Colony-forming assays confirmed the synergistic anticancer effects of CBD and GA when delivered via the MOF. Flow cytometry and immunofluorescence analyses indicated increased ROS production and changes in pro-apoptotic and anti-inflammatory markers, suggesting that the CBD/Mg-GA MOF induces apoptosis through mitochondrial dysfunction and the modulation of inflammation. Overall, the CBD/Mg-GA MOF showed promise as a dual drug carrier for GBM therapy, providing enhanced anticancer effects through the simultaneous delivery of CBD and GA. Further pre-clinical studies are recommended to confirm its safety and efficacy for cancer treatment [63].

## 6. MOFs in the Treatment of Parkinson

Parkinson’s disease (PD) is a progressive neurological disorder marked by motor dysfunction resulting from the degeneration of dopamine-producing neurons in the brain. Liu et al. investigated a novel approach by developing a ZIF-8@PB nanocomposite, where ZIF-8 was coated with Prussian blue (PB), forming particles with an average size of 107 nm and loaded with quercetin (QCT), a natural antioxidant aimed at treating PD (Figure 2a). This ZIF-8@PB-QCT nanocomposite demonstrated a strong response to near-infrared (NIR) light. The system utilizes PB’s photothermal effect to enhance BBB permeability, allowing QCT to enter neuronal cells and exert therapeutic effects directly in response to mitochondrial damage (Figure 2b). The drug release was assessed over a 24 h period, both with and without exposure to an 808 nm NIR laser. In the absence of the laser, only 13% of QCT was released, but with NIR laser application, around 77.96% of QCT was released from the ZIF-8@PB-QCT complex. This increased release was due to the heating effect, which enhanced the Brownian motion and led to the dissociation of QCT from the nanocomposite. The ZIF-8@PB-QCT nanocomposite significantly improved mitochondrial function and reduced PD symptoms in mouse models by increasing the ATP levels, decreasing oxidative stress, and repairing dopaminergic neuronal damage. These effects were attributed to QCT’s neuroprotective properties, including its anti-inflammatory, antioxidant, and anti-apoptotic actions, which were further amplified by the nanocomposite’s effective drug encapsulation and NIR-triggered release. The underlying mechanism involves the activation of the PI3K/Akt signaling pathway, highlighting the potential of this biocompatible, NIR-responsive system for treating neurodegenerative diseases by overcoming BBB barriers and targeting mitochondrial dysfunction [64].

In a distinct study, Jiang et al. [65] introduced an innovative approach to addressing PD by utilizing nanozyme-integrated MOFs to target neuroinflammation. The researchers developed chiral nanozymes by incorporating ultra-small platinum nanozymes (Ptzymes) into L-chiral and D-chiral imidazolate zeolite frameworks, resulting in Ptzyme@L-ZIF and Ptzyme@D-ZIF. Notably, the Ptzyme@D-ZIF variant exhibited enhanced brain accumulation in male PD mouse models, attributed to its extended plasma retention and the ability to cross the BBB via multiple mechanisms, including clathrin- and caveolae-mediated endocytosis. This led to significant improvements in reducing behavioral disorders and pathological changes. Moreover, Ptzyme@D-ZIFs demonstrated superior reactive oxygen species (ROS) scavenging abilities compared to Ptzyme@L-ZIFs due to their longer in vivo half-life and increased brain accumulation. Given the critical role of the BBB in both the development and treatment of brain diseases by regulating substance access to the brain, Ptzyme@D-ZIFs’ superior antioxidant activity, effective BBB penetration, and enhanced brain accumulation present them as promising candidates for combating ROS and inflammation. Both in vitro and in vivo assessments confirmed that Ptzyme@D-ZIFs effectively prevented neuroinflammation-induced apoptosis and ferroptosis in damaged neurons. The cytotoxicity of Ptzyme@D-ZIFs was assessed using an MTT assay, revealing minimal toxicity to SH-SY5Y cells at concentrations below 80 μg/mL, a dosage selected for further experimentation. Furthermore, Ptzyme@D-ZIFs showed a protective effect against MPP+-induced cell death, restoring cell viability to over 85% of the control group’s level. This study highlights the potential of nanozyme-based chiral nanomaterials in treating PD through differential metabolism and multi-mechanism therapies, paving the way for new drug development and pharmaceutical applications targeting brain diseases [65].

A research team in China, Li et al., explored the potential of the MOF@Man liposome nanozyme system for PD treatment by targeting the brain and enhancing therapeutic effects. This nanozyme, constructed from a nanoscale MOF (Zr-FeP MOF) that combines Fe-5,10,15,20-tetra (4-carboxyphenyl) porphyrin (Fe-TCPP) with Zr6 clusters, was encapsulated in mannitol (Man)-coated liposomes to improve BBB permeability. The system demonstrated significant brain accumulation, especially in the substantia nigra pars compacta and striatum, and effectively addressed issues such as oxidative stress, mitochondrial dysfunction, and neuroinflammation in MPTP-induced PD mouse models. Behavioral assessments showed enhanced motor function and memory in the treated mice, and a transcriptomic analysis revealed the normalization of critical PD-related genes like Parkin, Snca, Pink1, and Th. The MOF@Man liposome exhibited favorable biosafety with no adverse effects on major organs or liver and kidney functions. This study highlights the MOF@Man liposome’s potential for targeted PD therapy, emphasizing its capability to cross the BBB, modulate neuroinflammation and oxidative stress, and alleviate PD symptoms [66].

## 7. MOFs in the Treatment of Alzheimer’s

Amyloid β (Aβ) deposition is a key pathological hallmark of Alzheimer’s disease (AD), making it crucial to inhibit Aβ aggregation and disaggregate Aβ fibrils for effective treatment. In this study, Yang and colleagues developed a novel gold nanoparticle-enhanced MIL-101(Fe) (AuNPs@PEG@MIL-101) designed to inhibit Aβ. The framework’s high positive charge facilitated the binding and aggregation of Aβ40 on the nanoparticle surfaces. The gold nanoparticles (AuNPs) improved the surface properties of MIL-101, allowing for more uniform interaction with Aβ monomers and fibrils. This composite successfully inhibited the formation of extracellular Aβ monomer fibrils and disrupted existing Aβ fibrils. Additionally, AuNPs@PEG@MIL-101 reduced intracellular Aβ40 aggregation and lowered the amount of Aβ40 on the cell membrane, thereby protecting PC12 cells from Aβ40-induced damage to microtubules and cell membranes. Characterization confirmed the successful incorporation of AuNPs into MIL-101, enhancing its biocompatibility and distribution. Functional assays showed that AuNPs@PEG@MIL-101 was more effective in inhibiting and disaggregating Aβ40 fibrils compared to PEG@MIL-101 alone. Moreover, this composite protected PC12 cells from Aβ40-induced toxicity by preventing apoptosis and reducing ROS levels. It also decreased the cellular uptake of Aβ40 and its accumulation on cell membranes, mitigating associated cytotoxic effects. Finally, AuNPs@PEG@MIL-101 preserved cytoskeletal integrity by reducing Aβ40-induced microtubule damage. Overall, AuNPs@PEG@MIL-101 demonstrates considerable potential as a therapeutic agent for AD, offering an innovative approach to modulating Aβ aggregation and protecting neuronal cells from Aβ-induced damage [67].

In a recent study, Fe-MIL-88B-NH_2_, a magnetic nanomaterial, was used to encapsulate methylene blue (MB), a tau aggregation inhibitor, for diagnosing and treating AD [68]. As shown in Figure 3a, the Fe-MIL-88B-NH_2_ surface was functionalized with 1,4,7-triazacyclononane-1,4,7-triacetic acid (NOTA) and 5-amino-3-(pyrrolo [2,3-c]pyridin-1-yl)isoquinoline (defluorinated MK6240, DMK6240) to improve the targeting of hyperphosphorylated tau proteins, resulting in Fe-MIL-88B-NH_2_-NOTA-DMK6240/MB. The MB release from Fe-MIL-88B-NH_2_-NOTA-DMK6240/MB was gradual, decreasing over time in phosphate-buffered saline (PBS), with significant release observed within the first 2 h, followed by a slower release rate (Figure 3b). The relaxation value of Fe-MIL-88B-NH_2_-NOTA-DMK6240 correlated with the iron concentration, affecting the contrast in images. In a rat model of AD, induced by okadaic acid (OA) to promote tau protein hyperphosphorylation, Fe-MIL-88B-NH_2_-NOTA-DMK6240/MB effectively inhibited tau protein hyperphosphorylation and aggregation, reduced neuronal death in the hippocampus, and enhanced cognitive function. Cytotoxicity tests in SH-SY5Y cells using the MTT assay showed that 83% of cells remained viable after 24 h of exposure to 100 μg/mL of Fe-MIL-88B-NH_2_-NOTA-DMK6240, with viability dropping to 71% when MB was included. Furthermore, histological analyses of rat brains and major organs following hippocampal injection revealed no significant morphological changes, confirming the safety of these nano MOF materials in normal tissues (Figure 3c,d). Research on targeting tau pathology in AD continues to focus on inhibiting tau protein phosphorylation and aggregation to prevent their spread between cells [68].

As illustrated in Figure 4a, Wang et al. [69] explored a nanoparticle-based photooxygenation method using porphyrinic Zr MOF PCN-224 (PCN: porous coordination network) nanoparticles to inhibit Aβ aggregation with NIR light. These PCN-224 nanoparticles, which incorporate tetra-kis(4-carboxyphenyl)porphyrin (TCPP) ligands into a porous MOF structure, exhibited high singlet oxygen generation, excellent biocompatibility, and strong stability in physiological environments. Upon activation by NIR light, the nanoparticles effectively prevented the aggregation of Aβ monomers into β-sheet structures and significantly reduced Aβ-induced cytotoxicity. The PCN-224 nanoparticles were characterized by a stable crystalline framework, good dispersibility, and high photostability. An analysis using Thioflavin T (ThT) assays, circular dichroism (CD) spectra, and atomic force microscopy (AFM) confirmed their ability to inhibit Aβ42 aggregation under NIR light. Additionally, PCN-224 nanoparticles showed superior singlet oxygen production compared to traditional photosensitizers like rose bengal (RB) (Figure 4b).

In vitro studies with PC12 cells revealed that photoactivated PCN-224 nanoparticles notably decreased Aβ-induced cytotoxicity, highlighting their potential for treating AD. This NIR light-induced photooxygenation strategy offers several advantages over traditional methods, including enhanced water dispersity, deeper brain penetration, and improved stability and efficiency. Consequently, this approach holds significant promise for the non-invasive phototreatment of neurodegenerative diseases like Alzheimer’s [69].

Researchers introduced an innovative nanodrug delivery system, TR-ZRA, which was camouflaged with an erythrocyte membrane to effectively target the BBB and enhance the immune environment in AD. This system employed an MOF, Zn-CA, loaded with a CD22 shRNA plasmid to reduce CD22 expression in aging microglia. This reduction improved the microglia’s ability to clear Aβ and decreased complement activation, leading to increased neuronal activity and reduced inflammation in the AD brain. TR-ZRA also contained Aβ aptamers, allowing for the cost-effective monitoring of Aβ plaques in vitro. The system’s design incorporated transferrin receptor aptamers, enabling it to cross the BBB and effectively target the brain. Both in vivo and in vitro studies showed that TR-ZRA was safe and could modulate complement levels through chlorogenic acid, which further reduced immune inflammatory damage. The combined effects of chlorogenic acid and CD22 shRNA improved the microglial phagocytosis of Aβ, lessened the Aβ burden, inhibited neuroinflammation, and enhanced memory function in AD models. These results indicated that TR-ZRA provided a promising approach to AD therapy and held potential for application in other neurodegenerative diseases. The system’s comprehensive design not only addressed the immune microenvironment in AD but also offered a novel method for observing Aβ plaques, showcasing its extensive modulatory capabilities and potential in tackling brain aging and neurodegenerative disorders [70].

Wang et al. [71] developed and evaluated porphyrinic MOF and indocyanine green (PCN−222@ICG) nanoprobes designed to tackle AD by improving BBB penetration and enhancing therapeutic efficacy through photothermal and photo-oxygenation effects. The researchers modified PCN−222@ICG nanoprobes with rabies virus glycoprotein (RVG) peptides to improve their ability to cross the BBB. They employed a microfluidic brain-on-a-chip model, which simulated the human brain environment and demonstrated that RVG-modified nanoprobes achieved approximately 2.6 times higher BBB permeability compared to non-RVG-modified counterparts. This enhancement was also confirmed using a static BBB Transwell model, which showed consistent results. The study investigated the ability of these nanoprobes to target and disassemble Aβ plaques in ex vivo brain slices from AD mouse models. NIR-activated PCN−222@ICG@RVG nanoprobes significantly reduced the density and size of Aβ plaques compared to treatments with NIR alone or with PCN−222@RVG without NIR. A quantitative analysis using Thioflavin S staining revealed a substantial decrease in plaque formation in samples treated with NIR-activated PCN−222@ICG@RVG. The findings highlight the effectiveness of combining photothermal and photo-oxygenation effects in inhibiting Aβ42 aggregation and reducing neurotoxicity. The study concluded that using PCN−222@ICG@RVG nanoprobes holds promise as a therapeutic strategy for AD by mitigating the adverse effects of Aβ plaques. However, the authors acknowledged limitations such as the lack of precise measurements of Aβ aggregation and the need for in vivo studies to further validate the potential of these nanoprobes [71].

## 8. MOFs in the Treatment of Stroke

Ischemic stroke involves the loss of neural cells due to a disruption in blood flow, leading to significant brain damage. Neural stem cell (NSC) therapy offers a promising approach for addressing stroke-related damage beyond this critical window, potentially aiding in the replacement of lost and damaged neurons. In this research area, the potential of calcium-based MOFs (Ca-MOFs) conjugated with miR-124 was explored to enhance NSC therapy for ischemic stroke. Yang et al. [72] utilized NSC therapy as a potential treatment for stroke, addressing the issue of low NSC differentiation by employing neuron-specific miR-124 to advance the maturation of NSCs into functional neurons. To address the instability and low internalization of miR-124, they introduced a Ca-MOF-based nano delivery system (Figure 5a). The Ca-MOF@miR-124 nanoparticles were designed to improve the delivery and stability of miR-124, aiming to enhance neuronal differentiation (Figure 5b). The in vitro results reveal that these nanoparticles significantly increased neuronal differentiation compared to the controls, with differentiated NSCs exhibiting higher expression levels of neuronal markers Tuj1 and MAP2, longer neurites, and increased synaptic complexity. Functional neuronal activity was confirmed through multielectrode array recordings, which demonstrated robust spike activity indicative of mature neuronal function. In vivo evaluations using a mouse model of ischemic stroke showed that Ca-MOF@miR-124 nanoparticles, in combination with NSC transplantation, markedly reduced infarct size and improved neuronal survival. Histological analyses and neurological assessments indicated significant recovery compared to other treatments. This study highlighted the effectiveness of Ca-MOF@miR-124 nanoparticles in enhancing NSC therapy by promoting neuronal differentiation and functional integration, offering a promising strategy for advancing stroke treatment and potentially other neurodegenerative disorders [72].

The pathophysiology of ischemic stroke involves a complex interplay of mechanisms, with oxidative stress playing a central role in secondary injury. Following the initial ischemic event, reperfusion can lead to the excessive production of ROS such as superoxide anion, hydrogen peroxide, and hydroxyl radicals. These ROS contribute significantly to secondary damage in the central nervous system by inducing immune responses, inflammation, and ischemia–reperfusion injuries [73]. He and coworkers explored the potential of CeO_2_@ZIF-8 (CeO_2_: cerium dioxide) nanoparticles in mitigating ischemic stroke using an MCAO rat model. CeO_2_@ZIF-8 was synthesized to combine the antioxidant properties of CeO_2_ with the peroxidase activity of ZIF-8, aiming to enhance its therapeutic efficacy. In vitro transwell assays demonstrated CeO_2_@ZIF-8’s high permeability across the BBB and its internalization by PC12 cells through lysosome-mediated endocytosis. In vivo evaluations revealed that treatment with CeO_2_@ZIF-8 led to a significant reduction in the infarct size and an improvement in the neurological scores compared to the saline controls. Detailed analyses showed that CeO_2_@ZIF-8 effectively decreased superoxide anions, malondialdehyde (MDA), and inflammation markers (TNF-α, IL-1β, and IL-6), while preserving or enhancing antioxidative enzyme activities (SOD and GSH-Px). The nanosystem was well tolerated, with no apparent toxicity being observed in major organs and minimal adverse effects. Pharmacokinetic studies indicated a prolonged circulation time and enhanced brain accumulation of CeO_2_@ZIF-8 compared to free CeO_2_. Histological examinations further confirmed the reduction in neuronal damage and apoptosis. These findings highlight CeO_2_@ZIF-8’s dual functionality in both scavenging ROS and suppressing neuroinflammation, presenting it as a promising candidate for ischemic stroke therapy. This study offered a novel approach by integrating ZIF-8 with CeO_2_ for improved therapeutic outcomes and safety in stroke management. Researchers have also explored the theranostic applications of MOF-based materials for brain disorders [74].

In cases of brain damage resulting from ischemic stroke, a new therapeutic target is being proposed. When an ischemic stroke occurs, the sudden cessation of blood flow causes an excessive release of glutamate, leading to the overactivation of N-methyl-D-aspartate receptors (NMDARs). This overstimulation activates neuronal nitric oxide synthase (nNOS), which then produces excessive nitric oxide (NO), further aggravating brain damage. The researchers recommend focusing on the NMDAR-postsynaptic density protein-95 (PSD-95)-nNOS signaling pathway as a potential avenue for developing more effective treatments [75].

Researchers came up with a novel way to screen PSD95-nNOS uncouplers quickly and sensitively using magnetic Fe-MOF. Layer-by-layer self-assembly was used to construct these novel PSD95-nNOS/Fe-MOF nanoparticles. First, His-nNOS was immobilized on magnetic Fe-MOF and then GFP. The system’s practicality was increased by the MMOF nanoparticles, which were extensively characterized and showed excellent stability, anti-interference capabilities, and recyclability. Using this method, screen candidates for PSD95-nNOS uncoupling were found, including Baicalin, Baicalein, Gnetol, Emodin-8-O-β-D-glucopyranoside, and Corylifol A. These candidates were then subjected to additional research. Using multifunctional PSD95-nNOS/Fe-MOF nanoparticles, this innovative approach provides a useful in vitro screening model to identify putative PSD95-nNOS uncouplers. A novel technique for quickly and sensitively screening putative uncouplers was created by immobilizing His-nNOS and GFP-PSD95 on the surface of magnetic Fe-MOF. The biocomposites showed outstanding stability, anti-interference, and reusability after they were made successfully. After interacting with the uncouplers, the MMOF nanoparticles could be readily isolated from uncoupling PSD95 using a magnet. Fluorescence intensities were then utilized to assay the uncoupling efficiency at a high throughput level. Natural actives were successfully identified using this screening approach as possible PSD95-nNOS uncouplers. To sum up, the creation of PSD95-nNOS/Fe-MOF nanoparticles as a novel material and process will greatly aid in the high throughput screening process’s ability to find PSD95-nNOS uncouplers [76].

At the conclusion of this review, we present a comprehensive summary of the key details pertaining to the MOFs discussed, as outlined in Table 2. This table elegantly consolidates essential information about the various MOFs, emphasizing their mechanisms for drug delivery, approaches to traversing the BBB, targeted diseases, and their therapeutic efficacy.

In summary, using MOFs effectively to overcome the BBB in the treatment of various neurological diseases involves employing diverse strategies such as mimicking viral designs and utilizing peptides, magnetic nanoparticles, and light activation. These MOF-based systems have demonstrated improved therapeutic efficacy for multiple disease types, including glioblastoma, Parkinson’s disease, Alzheimer’s disease, and ischemic stroke. The ability of these MOF systems to enhance drug delivery, reduce disease-related symptoms, and improve neurological outcomes underscores their potential as a promising approach in the field of brain-targeted therapies.

## 9. Challenges in Developing MOFs for Treating Brain Diseases

Developing MOFs for treating brain diseases involves navigating a complex set of challenges, especially regarding the BBB, long-term safety, metabolic pathways, clinical translation, and high-cost productions. Despite their promising potential, MOFs face considerable obstacles as they move from laboratory research to clinical application. One primary challenge is enabling MOFs to penetrate the BBB, a critical defense mechanism that safeguards the brain from potentially harmful substances. Designing MOFs to cross this barrier while preserving its protective function requires meticulous engineering to ensure effective delivery and minimal disruption to the BBB’s integrity.

Ensuring the long-term biosafety of MOFs in the brain presents another significant challenge. Given the brain’s heightened sensitivity to foreign substances, it is crucial that the metal ions and organic linkers used in MOFs are non-toxic and compatible with neural tissue. Furthermore, MOFs must retain their structural stability and drug-carrying capacity within the brain’s complex physiological environment, which includes fluctuating pH levels and varying enzymatic activity. Achieving controlled drug release in response to specific brain conditions or external stimuli is essential but technically demanding. Precision targeting of specific brain regions or cell types, minimizing off-target effects, and maintaining consistent nanoscale dimensions are vital for effective treatment. Additionally, MOFs must be biodegradable after their therapeutic function to prevent long-term accumulation and associated side effects. Recent studies underscore the need to address the neurotoxicity of MOFs. Research has shown that certain MOFs, such as ZIF-67, can induce neurotoxic effects, including impaired memory and learning functions in animal models, highlighting the need to develop MOFs with low or non-neurotoxic profiles. Further research is necessary to understand and mitigate the toxic effects of MOFs on neurons and other brain cells [77].

The metabolism and excretion of MOFs after their uptake by neural cells also present critical considerations. The transformation of MOFs within the central nervous system can influence their physiological effects and therapeutic efficacy. A detailed understanding of how MOFs are metabolized and eliminated from the brain is essential for ensuring their safety and effectiveness. For example, MOFs designed for glioma-targeted drug delivery have shown promise in animal models, but their absorption, metabolism, and excretion remain unexplored. Research on a metalloporphyrin-integrated nanosystem (MOF@MP-RGD) following intrathecal administration revealed that this nanosystem accumulates in brain tumor tissue, is metabolized by the liver, and is predominantly excreted through feces. This metabolic pathway reduces long-term retention and potential in vivo toxicity, supporting the clinical feasibility of the treatment [78].

The clinical translation of MOF-based therapies faces challenges, including the complexity of MOF structures, scalability and reproducibility issues, and stringent regulatory requirements. The transition from preclinical research to clinical trials is both time-consuming and expensive, with rigorous approval processes adding to the complexity. Overcoming these challenges will require collaborative efforts between materials scientists, pharmaceutical researchers, and regulatory agencies to develop safer and more effective MOF-based therapies and facilitate their progression into clinical practice.

Finally, the high production costs and complex synthesis processes of MOFs pose other challenges in treating brain diseases. Their intricate preparation and poor stability under physiological conditions add to the expense and restrict their practical applications. Addressing these issues requires the development of simple, cost-effective, and environmentally friendly synthesis methods, such as green techniques, to improve the feasibility and scalability of MOFs in clinical settings.

## 10. Conclusions

In conclusion, recent advancements in MOFs have introduced promising approaches for treating brain diseases, including glioblastoma multiforme, Parkinson’s disease, Alzheimer’s disease, and ischemic stroke. The development of novel nanoplatforms has shown significant efficacy in inducing targeted cell death in tumors, enhancing drug delivery to the brain, and reducing neuroinflammation and oxidative stress. MOFs utilize various mechanisms, such as inducing specific forms of cell death, improving BBB permeability, and targeting disease-specific proteins. Their ability to combine therapeutic and diagnostic functions presents a new frontier in neurodegenerative disease management. Overall, these innovations hold great potential for advancing treatment strategies and improving patient outcomes in complex brain disorders.

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
