# Peer review of "Metal–Organic Frameworks for Overcoming the Blood–Brain Barrier in the Treatment of Brain Diseases: A Review"

_nanomaterials, 2024, doi:10.3390/nano14171379_

Round 1
Reviewer 1 Report
Comments and Suggestions for Authors
This review by Rohani et al provides a comprehensive overview of the use of metal-organic frameworks (MOFs) in overcoming the blood-brain barrier (BBB) for the treatment of various brain diseases. ​ The paragraphing is concise and easy to understand. I have the following suggestions and questions related to this work. Addressing these questions could further enrich the study findings and provide a more comprehensive understanding of the applications and limitations of MOFs for treating BBB related disease treatment.
1.Long-Term Safety: This review discusses the efficacy and potential of MOFs in treating brain diseases, but it does not delve into the long-term safety implications of using MOFs for drug delivery to the brain. ​ Understanding the potential risks, such as accumulation in the brain or systemic toxicity over extended periods, is crucial for assessing the overall safety profile of MOFs. ​It is very important to cite relevant literature that studied the long-term effects of MOFs on brain and the mechanisms by which the MOFs are excreted out of the body.
2. Clinical Translation: Though the authors have precisely described about the importance of scalability and reproducibility in MOF synthesis for clinical translation, the review does not elaborate on the current status of MOF-based treatments in terms of clinical trials or regulatory approval. ​ Providing insights into the progress of MOF research from the laboratory to clinical applications would offer a more comprehensive view of the field. ​Why there is so much research in the field of nanoscience but very negligible amount of the researched compounds is entering clinical trial stage.
3. Comparison with Other Nanoparticle Systems: Although the authors highlight the advantages of MOFs in drug delivery to the brain, they do not compare MOFs with other nanoparticle-based delivery systems mentioned earlier in the text, such as liposomes, micelles, and nanoparticles. ​ A comparative analysis of MOFs against existing systems in terms of efficacy, safety, and scalability would provide a more nuanced understanding of MOFs' unique contributions. ​I suggest the authors to tabulate the data and compare the advantages and limitations of each class of nanoparticle.
4. Cost and Accessibility: The review does not address the cost implications or accessibility of MOF-based treatments for brain diseases. Understanding the economic feasibility of using MOFs in clinical settings, as well as their availability for widespread use, is essential for evaluating their potential impact on healthcare systems and patient access to innovative treatment.
Author Response
Response to Reviewer 1
Comments and Suggestions for Authors
This review by Rohani et al provides a comprehensive overview of the use of metal-organic frameworks (MOFs) in overcoming the blood-brain barrier (BBB) for the treatment of various brain diseases. ​ The paragraphing is concise and easy to understand. I have the following suggestions and questions related to this work. Addressing these questions could further enrich the study findings and provide a more comprehensive understanding of the applications and limitations of MOFs for treating BBB related disease treatment.
- Long-Term Safety: This review discusses the efficacy and potential of MOFs in treating brain diseases, but it does not delve into the long-term safety implications of using MOFs for drug delivery to the brain. ​ Understanding the potential risks, such as accumulation in the brain or systemic toxicity over extended periods, is crucial for assessing the overall safety profile of MOFs. ​It is very important to cite relevant literature that studied the long-term effects of MOFs on brain and the mechanisms by which the MOFs are excreted out of the body.
Answer: Thank you for the comments. Based on your suggestion, we have added a section discussing the challenges in the revised manuscript on page 31.
- Clinical Translation: Though the authors have precisely described about the importance of scalability and reproducibility in MOF synthesis for clinical translation, the review does not elaborate on the current status of MOF-based treatments in terms of clinical trials or regulatory approval. ​ Providing insights into the progress of MOF research from the laboratory to clinical applications would offer a more comprehensive view of the field. ​Why there is so much research in the field of nanoscience but very negligible amount of the researched compounds is entering clinical trial stage.
Answer: Thank you for your feedback. We have added the following paragraph to the section discussing the challenges on page 32.
After searching, we couldn't find any instances where MOFs have been used from laboratory research to clinical applications. The paper by Tayagi and colleagues [1] examines the slow transition of MOFs from laboratory research to clinical applications, despite significant interest and research. The review highlights several key factors impeding the advancement of MOF-based treatments into clinical trials. A major challenge is the complexity and variability of MOF structures, which complicates the prediction of their toxicity, stability, and biocompatibility. Differences in size, shape, and metal content can significantly impact their behavior, making it difficult to standardize safe and effective formulations. The review also points out that the instability of MOFs in biological fluids, due to the kinetic lability of metal centers, can lead to premature degradation and reduced effectiveness. Additionally, the limited exploration of alternative administration routes, such as local or non-invasive methods, poses further challenges. The stringent regulatory requirements for new drug approvals add to the difficulty, making the process time-consuming and costly. In contrast, medical devices incorporating MOFs face less rigorous approval processes. The review concludes that addressing these challenges through targeted research and collaboration is crucial for advancing MOF-based treatments into clinical practice. Finally, I mentioned these limitations in the section titled “Challenges in Developing MOFs for Treating Brain Diseases.”
- Tyagi, Y.H. Wijesundara, J.J. Gassensmith, et al., Clinical translation of metal–organic frameworks, Nat Rev Mater, 8 (2023), pp. 701–703.
- Comparison with Other Nanoparticle Systems: Although the authors highlight the advantages of MOFs in drug delivery to the brain, they do not compare MOFs with other nanoparticle-based delivery systems mentioned earlier in the text, such as liposomes, micelles, and nanoparticles. ​ A comparative analysis of MOFs against existing systems in terms of efficacy, safety, and scalability would provide a more nuanced understanding of MOFs' unique contributions. ​I suggest the authors to tabulate the data and compare the advantages and limitations of each class of nanoparticle.
Answer: Thank you for your feedback. We have incorporated your suggestions on pages 3 and 4 and compared the advantages and disadvantages of MOFs with other nanoparticle-based delivery systems, such as liposomes, micelles, and nanoparticles in Table 1.
- Cost and Accessibility: The review does not address the cost implications or accessibility of MOF-based treatments for brain diseases. Understanding the economic feasibility of using MOFs in clinical settings, as well as their availability for widespread use, is essential for evaluating their potential impact on healthcare systems and patient access to innovative treatment.
Answer: Thank you for your comment. We have incorporated the additions you suggested in the section addressing the challenges on page 32.

Reviewer 2 Report
Comments and Suggestions for Authors
Overall, this manuscript is well-structured and well-written. It is a review manuscript that exceeds the average level of MDPI. I suggest that the following revisions be made before publication.
Major:
1. The number of articles on MOFs applied straightly to overcome BBB listed by the author is too less. From ref 59-73 (excluding 60, 70, 72), there are only 11 articles in total (although these articles are fresh). All other references are background introductions. It is recommended to greatly increase the content of this part, at least citing and introducing 20-30 recent high-level research papers.
2. I’d recommend the authors set up a table in Part 4, including the name of MOFs, the drug delivered, how to overcome BBB, disease type, therapeutic efficiency, et. al from all the corresponding reference the authors introduced in this part. In addition, I’d recommend to set up subtitles in Part 4, for example, “4.1 MOFs in treatment of GBM”, and so on.
3. Lacking of figures in Stroke therapy part.
Minor:
1. Many paragraphs are too long and uncomfortable to read. I recommended to reduce the content as possible or divide them into several paragraphs.
2. The Scheme 1 drawn by the authors is beautiful. The only problem is that the colors are too bright and the saturation is too high. If the colors can be darker, it will have a better aesthetic effect.
3. In scheme 1, “Characterization” and “synthesis” are reversed.
4. Delete other “(BBB)”, “(MOFs)”, “(5-FU)”,”(MILs)”,”(GBM)” excluding the first appearance in text. Some other abbreviations also appear in wrong place such as “(ZIF-8)”. Please check all abbreviations carefully and make a list of them.
5. “In vivo” and “in vitro” should be italic.
Author Response
Response to Reviewer 2
Comments and Suggestions for Authors
Overall, this manuscript is well-structured and well-written. It is a review manuscript that exceeds the average level of MDPI. I suggest that the following revisions be made before publication.
Major:
- The number of articles on MOFs applied straightly to overcome BBB listed by the author is too less. From ref 59-73 (excluding 60, 70, 72), there are only 11 articles in total (although these articles are fresh). All other references are background introductions. It is recommended to greatly increase the content of this part, at least citing and introducing 20-30 recent high-level research papers.
Answer: Thank you very much for your valuable feedback. We were not able to locate additional articles specifically discussing metal-organic frameworks for overcoming the BBB in the treatment of brain diseases. We did find and included three other relevant papers in the revised manuscript on pages 15, 23, and 24.
- I’d recommend the authors set up a table in Part 4, including the name of MOFs, the drug delivered, how to overcome BBB, disease type, therapeutic efficiency, et. al from all the corresponding reference the authors introduced in this part. In addition, I’d recommend to set up subtitles in Part 4, for example, “4.1 MOFs in treatment of GBM”, and so on.
Answer: Thanks for your valuable comment. We have included them as per your recommendation on pages 27 to 30.
- Lacking of figures in Stroke therapy part.
Answer: Thank you very much for the comment. We have added it as Figure 5.
Top of Form
Bottom of Form
Minor:
- Many paragraphs are too long and uncomfortable to read. I recommended to reduce the content as possible or divide them into several paragraphs.
Answer: Thank you so much for your insightful comments that helped improve the article. We have separated some paragraphs in both the Introduction and the Challenges sections.
- The Scheme 1 drawn by the authors is beautiful. The only problem is that the colors are too bright and the saturation is too high. If the colors can be darker, it will have a better aesthetic effect.
Answer: Thank you for your feedback. We have revised Scheme 1.
- In scheme 1, “Characterization” and “synthesis” are reversed.
Answer: Thanks for the comment. We have revised it.
- Delete other “(BBB)”, “(MOFs)”, “(5-FU)”,”(MILs)”,”(GBM)” excluding the first appearance in text. Some other abbreviations also appear in wrong place such as “(ZIF-8)”. Please check all abbreviations carefully and make a list of them.
Answer: Thank you for your comment. We have revised.
- “In vivo” and “in vitro” should be italic.
Answer: Thank you for your comment. We have revised.

Round 2
Reviewer 1 Report
Comments and Suggestions for Authors
I do not have any further concerns. The authors have addressed all the questions that I've raised.
Reviewer 2 Report
Comments and Suggestions for Authors
The authors solved all my concerns.